# Modulation of the MOP Receptor (μ Opioid Receptor) by Imidazo[1,2-*a*]imidazole-5,6-Diones: In Search of the Elucidation of the Mechanism of Action

**DOI:** 10.3390/molecules27092930

**Published:** 2022-05-04

**Authors:** Dominik Straszak, Agata Siwek, Monika Głuch-Lutwin, Barbara Mordyl, Marcin Kołaczkowski, Aldona Pietrzak, Mansur Rahnama-Hezavah, Bartłomiej Drop, Dariusz Matosiuk

**Affiliations:** 1Department of Synthesis and Chemical Technology of Pharmaceutical Substances, Faculty of Pharmacy, Medical University, Chodzki 4A, 20-093 Lublin, Poland; dominik.straszak@gmail.com; 2Faculty of Pharmacy, Jagiellonian University Medical College, Medyczna 9, 30-688, Kraków, Poland; agat.siwek@uj.edu.pl (A.S.); monika.gluch@uj.edu.pl (M.G.-L.); barbara.mordyl@uj.edu.pl (B.M.); marcin.kolaczkowski@uj.edu.pl (M.K.); 3Department of Dermatology, Venereology, and Paediatric Dermatology, Faculty of Medicine, Medical University, Staszica 11, 20-080 Lublin, Poland; aldona.pietrzak@umlub.pl; 4Chair and Department of Oral Surgery, Medical University, Chodzki 6, 20-093 Lublin, Poland; mansur.rahnama@umlub.pl; 5Department of Medical Informatics and Statistics, Medical University, Jaczewskiego 4, 20-090 Lublin, Poland; bartlomiej.drop@umlub.pl

**Keywords:** MOP receptor, OP3 receptor, μ-opioid receptor ligands, allosterism, antagonism, imidazo[1,2-*a*]imidazoles

## Abstract

The μ-opioid receptors belong to the family of G protein-coupled receptors (GPCRs), and their activation triggers a cascade of intracellular relays with the final effect of analgesia. Classical agonists of this receptor, such as morphine, are the main targets in the treatment of both acute and chronic pain. However, the dangerous side effects, such as respiratory depression or addiction, significantly limit their widespread use. The allosteric centers of the receptors exhibit large structural diversity within particular types and even subtypes. Currently, a considerable interest is aroused by the modulation of μ-opioid receptors. The application of such a technique may result in a reduction in the dose or even discontinuation of classical opiates, thus eliminating the side effects typical of this class of drugs. Our aim is to obtain a series of 1-aryl-5,6(1*H*)dioxo-2,3-dihydroimidazo[1,2-*a*]imidazole derivatives and provide more information about their activity and selectivity on OP3 (MOP, human mu opioid receptor). The study was based on an observation that some carbonyl derivatives of 1-aryl-2-aminoimidazoline cooperate strongly with morphine or DAMGO in sub-threshold doses, producing similar results to those of normal active doses. To elucidate the possible mechanism of such enhancement, we performed a few in vitro functional tests (involving cAMP and β-arrestin recruitment) and a radioligand binding assay on CHO-K1 cells with the expression of the OP3 receptor. One of the compounds had no orthosteric affinity or intrinsic activity, but inhibited the efficiency of DAMGO. These results allow to conclude that this compound is a negative allosteric modulator (NAM) of the human μ-opioid receptor.

## 1. Introduction

Painkillers are one of the most commonly used drugs in the world. About 10% of the European population suffers from chronic pain [1]. The first step in the analgesic ladder is the use of a group of nonsteroidal anti-inflammatory drugs (NSAIDs). By blocking prostaglandin cyclooxygenase (COX), they exert anti-inflammatory, analgesic, and antipyretic effects. However, due to their relatively low efficacy in the treatment of severe pain and their side effects, including blood coagulation disorders, ototoxicity, nephrotoxicity, and damage to the stomach mucosa, quite often it is necessary to use stronger painkillers, such as opioids. Classical opiates currently used in medicine also have a number of side effects, which can be dangerous for health and life, such as respiratory depression, drowsiness, slow bowel motility, and ecstasy. Moreover, the tolerance developing during their use considerably limits their widespread use in practice. Therefore, there is a constant need for finding new substances that will induce strong analgesia and simultaneously present a better clinical safety profile. There is structural similarity between opioid receptors and other proteins from the GPRC family. MOP, DOP, and KOP have approximately 60% of the amino acid sequence in common, especially in the orthosteric site. This explains why it is so difficult to obtain compounds with a high receptor specificity. 

Lately, it has been shown that, in addition to the orthosteric activation of the opioid receptor, other routes are possible as well. Currently, many research programs focus on the allosteric modulation aspect, which increases the affinity and/or efficacy of orthosteric ligand that has the ability to activate the receptor [2]. The first compounds discovered were naturally occurring substances with modulating properties toward opioid receptors, such as cannabidiol, tetrahydrocannabinol (THC) [3], and salvinorin A [4]. They act as negative allosteric modulators (NAM) and, thus, weaken the effect of the agonist. In recent years, the first positive allosteric modulators (PAM) for MOPs have been obtained, i.e., BMS-986121, BMS-986122 [5], ignavine [6], and MS1 [7]. The findings of the research on PAM ligands were summarized in a review paper [8]. Other selective and dual acting ligands were reevaluated as well [9].

Another route of attenuation of the receptor is the biased agonism toward one of the possible signal transduction pathways—G-protein or β-arrestine. Biased agonism was confirmed for several MOP receptor ligands [10], and contributed to clinical trials, the marketing of oliceridine [11], and the discovery of PMZ21 [12]. 

Due to the high specificity of allosteric modulators to the receptors, new perspectives in pharmacotherapy emerged, aimed particular types and subtypes of receptors. At the same time, this eliminated the problem of side effects specific for classic opiates.

A number of imidazoline derivatives exerting an effect on the central nervous system (CNS) have been obtained in the Department of Synthesis and Technology of Drugs of the Medical University in Lublin for many years [13,14,15]. 1-aryl-5,6(1*H*)dioxo-2,3-dihydroimidazo[1,2-*a*]imidazoles showed particularly interesting properties, because their strong analgesic activity was confirmed in a series of behavioral tests (“writhing” and “hot plate” tests), and their pharmacological effect was only partly reversible when using naloxone. Moreover, they did not act depressively on the respiratory system [15], but, when given in small, sub-threshold doses, they cooperated positively enhancing the inactive doses of morphine or DAMGO [16]. In this paper, we present the results of the synthesis of 1-aryl-5,6(1*H*)dioxo-2,3-dihydroimidazo[1,2-*a*]imidazoles and their biological effect on the MOP (OP3) receptor, including functional tests.

## 2. Materials and Methods

### 2.1. Chemistry

Chemicals were purchased from Sigma-Aldrich (Saint Louis, MO, USA) and used without further purification. Melting points (m.p.) were measured using Boethius apparatus and are given uncorrected. ^1^H-NMR (600 MHz) was recorded on a Bruker Avance 600 apparatus (Bruker BioSpin GmbH, Rheinstetten, Germany) in DMSO-d_6_ with TMS (tetramethyl silane) as an internal standard. The atmospheric pressure chemical ionization (APCI) mass spectrometer microTOF-Q II (Bruker Daltonics, Bremen, Germany) was used for mass analysis. TLC was performed on Merck silica gel F_254_ plates with the eluent system: chloroform/methanol (4:1); visualization: UV light λ = 254. The purity of the compounds based on their H NMR spectra was over 95%.

### 2.2. General Procedure

A free base of 1-aryl-2-iminoimidazoline (0.1 mol) was added into excess of diethyl oxolate (200 mL) in an apparatus for distillation with Dean–Stark trap. The reaction was carried out in two temperatures ranges: at 100–120 °C for the first hour and at 200–220 °C for the next 30 min. The course of the reaction was measured by the amount of collected ethanol. The mixture was cooled for the next 24 h. The crude product was purified by recrystallization from propan-2-ol [14] (Figure 1). 

### 2.3. In Vitro Studies

#### Preparation of Solutions of the Tested and Reference Compounds

Stock solutions were prepared in the concentration of 10^−2^ M for the tested and reference compounds. A minimum 1 mg of each tested compound was weighed and dissolved in an appropriate volume of dimethyl sulfoxide. The solutions of the compounds were then stirred with a vortex and allowed to stand for 15 min in a water bath with ultrasound. Serial dilutions were prepared in phosphate-buffered saline (PBS) using the automated pipetting system epMotion 5070 (Eppendorf, Hamburg, Germany). Before assays, possible precipitation or opalescence was checked. For evaluation of the compounds’ affinity and intrinsic activity of the MOP, DAMGO, Endomorphin-1, Morphine, Codeine, CTOP, CTAP and β-FNA were used as a reference substance. All experiments were performed in duplicate in two independent experiments. 

### 2.4. Radioligand Binding Assay

Radioligand binding was performed using membranes from CHO-K1 cells stably transfected with the human μ-opioid receptor (PerkinElmer). A total of 50 μL working solution of the tested compounds, 50 μL [^3^H]-DAMGO (final concentration 0.5 nM), and 150 μL diluted membranes (5 μg protein per well) prepared in PBS (pH 7.4) were transferred to a polypropylene 96-well microplate using a 96-well pipetting station Rainin Liquidator (Mettler-Toledo, Columbus, OH, USA). DAMGO (10 μM) was used to define nonspecific binding. The microplate was covered with a sealing tape, mixed, and incubated for 60 min at 27 °C. The reaction was terminated by rapid filtration through GF/B filter mate presoaked with 0.5% polyethyleneimine for 30 min. Ten rapid washes with 200 μL 50 mM Tris buffer and 154 mM NaCl (4 °C, pH 7.4) were performed using a 96-well FilterMate harvester (PerkinElmer, Walham, MA, USA). The filter mates were dried at 37 °C in a forced air fan incubator and then solid scintillator MeltiLex was melted on the filter mates at 90 °C for 4 min. The radioactivity on the filter was measured in a MicroBeta TriLux 1450 scintillation counter (PerkinElmer, Walham, MA, USA). Data were fitted to a one-site curve-fitting equation with Prism 6 (GraphPad Software), and Ki values were estimated from the Cheng–Prusoff equation. 

### 2.5. Spectrum Absorbance

To verify the solubility of the compounds at a concentration of 10 μM and to exclude the precipitation of the compounds, the absorbance spectrum was measured in the range from 220 to 700 nm. The measurement was performed using a multifunctional plate reader PolarStar BMG LabTech. No precipitation of the tested compounds was observed. The spectrum of the compounds is shown in Appendix A.

### 2.6. Functional Assays

**cAMP Inhibition.** For the μ-opioid receptor, adenylyl cyclase activity was monitored using cryopreserved CHO-K1 cells with the expression of the human μ-opioid (OP3) receptor obtained from Perkin Elmer (Walham, MA, USA). Thawed cells were resuspended in stimulation buffer (HBSS, 5 mM HEPES, 0.5 IBMX, and 0.1% BSA at pH 7.4) at 2 × 10^5^ cells/mL. The same volume (10 μL) of the cell suspension was added to the tested compounds with 10 μM forskolin. The samples were loaded onto a white opaque 96-well half area microplate. The agonist assay was performed for a group of the five new compounds (**1a**, **1b**, **1c**, **1d**, **1e**) and the reference compounds in the concentration range from 10^−5^ M to 10^−12^ M. The antagonist response was determined at the same concentration range as in the agonist assay (10^−5^ M to 10^−12^ M). In this case, DAMGO was used in four concentrations, EC_20_ (1 nM), EC_50_ (4.6 nM), EC_80_ (18 nM), and EC_87_ (30 nM), as a reference agonist. The EC values were experimentally obtained from the curve of DAMGO and were calculated with GraphPad Software. The agonist was added simultaneously, but in the case of the antagonist assay, we performed preincubation with the compounds for 30 min before the reference agonist was added. After preincubation, the cells were stimulated for 30 min at room temperature (22 °C) and cAMP measurements were performed with a homogeneous TR-FRET immunoassay using the LANCE Ultra cAMP kit (PerkinElmer, Waltham, MA, USA). A total of 10 μL of Eu cAMP Tracer Working Solution and 10 μL of ULight-anti-cAMP Tracer Working Solution were added, mixed, and incubated for 1 h. The TR-FRET signal was read on an EnVision microplate reader (PerkinElmer, Waltham, MA, USA). Data were calculated with a three parameter dose–response curve (EC_50_) or dose–response curves of Inhibition (IC_50_) using Prism 6.0 (GraphPad Software).

**β-Arrestin Recruitment.** For the μ-opioid receptor, β-arrestin recruitment was monitored using cryopreserved U2OS cells with the expression of the human μ-opioid (OP3) receptor obtained from Life Technologies. This cell line contains the human Opioid Receptor Mu 1 linked to a TEV protease site and a Gal4-VP16 transcription factor stably integrated into the Tango GPCR-bla U2OS parental cell line. This parental cell line stably expresses a β-arrestin/TEV protease fusion protein and the beta-lactamase reporter gene under the control of a UAS response element. Thawed cells were resuspended in DMEM medium with high-glucose and GlutaMAX, 10% fetal bovine serum (FBS) charcoal stripped, 0.1 mM non-essential amino acids (NEAA), 25 mM HEPES, and penicillin 100 U/mL/streptomycin 100 μg/mL (antibiotics) at 3.125 × 10^5^ cells/mL onto a black view plate 384-well microplate. The cells to attach were incubated for 5 h. On the day of the assay, the serial dilution of the tested compounds (**1a**, **1b**, **1c**, **1d** and **1e**) and the reference compounds were added in the concentration range from 10^−5^ M to 10^−12^ M. The incubation was carried out for 16 h at 37 °C and 5% CO_2_. The agonist and antagonist were added simultaneously, but in the case of the antagonist assay, we performed preincubation with the compounds for 30 min before the reference agonist was added. DAMGO was used in two values, EC_20_ (1 nM) and EC_87_ (30 nM), as a reference agonist. The plates were incubated at 37 °C/5% CO_2_ for 5 h, after which time 8 μL of LiveBLAzer TR-FRET B/G substrate (Invitrogen, Waltham, MA, USA) with solution D was added per well to a final concentration of 1 μM. The plates were incubated in the dark at room temperature for 2 h and then read on FluoStar OPTIMA BMG Labtech (Ortenberg, Germany) with excitation at 410 nm and emission wavelengths at 440 nm and 530 nm.

Data were calculated with a three parameter dose–response curve (EC_50_) or dose–response curves of Inhibition (IC_50_) using Prism 6.0 (GraphPad Software). 

### 2.7. Reagents and Materials

[^3^H]-DAMGO, 250 μCi/mL, PerkinElmer, NET902250UC; 1/2 AREAPLATE-96, White, 199 μL volume, 1/2 well, Perkin Elmer, 6005560; 96 Polypropylene Microplate, U bottom, clear, Greiner Bio-One, 650 201; Calcium and magnesium-free PBS, Gibco, 14190144; cAMP LANCE Ultra cAMP Detection Kit, Perkin Elmer, TRF0263; cAMP Zen, Human μ-opioid (OP3) Receptor, Frozen Cells, Perkin Elmer, ES-542-CF; CTAP, Tocris, 1560; CTOP, Tocris, 1578; DAMGO, Sigma Aldrich, E7384; Dimethyl sulfoxide CZDA, ≥99.7%, POCh, 363550117; DMEM, Life Technologies, 10569; Elmer, 6007460; Endomorphin-1, Tocris, 1055; Ethylenediaminetetraacetic acid disodium salt dihydrate for mol. biol., ≥99%, Sigma-Aldrich, E5134; FBS dialyzed, Life Technologies, 26400044; Fetal Bovine Serum, charcoal stripped, USDA-approved regions, Life Technologies, 12676; Filtermat B, GF/B, PerkinElmer, 1450-442; Forskolin, Sigma Aldrich, F6886; Ham’s F-12 Nutrient Mix, GlutaMAX™ Supplement, Life Technologies, 31765; HEPES, Gibco, H0887; IBMX, Sigma Aldrich, I-5879; L-Glutamine, Gibco, 25030081; MeltiLex, for Microbeta Filters, PerkinElmer, 1450-441; Membrane Target System; human μ-opioid Receptor, Perkin Elmer, ES-542-M400UA; NEAA (non-essential amino acids), Gibco, 11140; Penicillin/Streptomycin (antibiotics), Life Technologies (Carlsbad, CA, USA), 15140; Sodium Pyruvate, Gibco, 25200072; Standard HBSS (with CaCl2 and MgCl2), Gibco, 14025; Trizma^®^ base BioPerformance Certified ≥99.0%, Sigma-Aldrich, T6066; Trizma^®^ hydrochloride BioPerformance Certified ≥99.0%, Sigma-Aldrich, T5941; Trypsin-EDTA, Gibco, 25200072; ViewPlate-384 Black, Optically Clear Bottom, Tissue Culture Treated, Sterile, 384-Well, Perkin; β-arrestin Tango™ OPRM1-bla U2OS DA Assay Kit, Life Technologies, K1599; β-Funaltrexamine hydrochloride (β-FNA), 0926.

## 3. Results

### 3.1. Radioligand Binding Assay

The radioligand binding experiments revealed diverse pKi values for the reference compounds. The highest pKi value was noted for endomorhin-1, whereas codeine had the lowest pKi. No binding of the tested compounds, **1a**, **1b**, **1c**, **1d** and **1e**, to the μ-opioid receptor was observed in any of the tested concentrations (10^−5^–10^−10^ M). Therefore, no calculations could be performed (Table 1). The dose–response curves for the radioligand binding assay of all the tested compounds are presented in the Appendix A.

Additionally, the affinity of DAMGO administered alone and in combination with the tested compounds was compared. The Ki value for DAMGO administered alone was 0.8 ± 0.03, and the addition of the tested compounds did not result in an increase in the affinity of DAMGO for the μ-opioid receptors. The results are shown in Table 2. The dose–response curves for the radioligand binding assay with DAMGO added to all the compounds are presented in the Appendix A. 

### 3.2. Intrinsic Activity of the Tested Compounds—An “In Vitro” Functional Assay

**cAMP Inhibition.** The results of the functional assays show varied E_max_ values of the reference substance. Three of the tested standards were classified as full agonists (the E_max_ percentage was similar to that of DAMGO). Codeine was classified as a partial agonist (E_max_ = 52%). In the intrinsic activity assays, no significant changes in cAMP levels were observed in the tested concentrations of all the tested compounds in the agonist and antagonist modes (Table 3). 

In the cAMP assay with the preincubation of the tested compounds for 30 min, compound 1c showed weak antagonist properties with E_max_% = 41 and pK_b_ 9.18 with 30 nM reference agonist of DAMGO. A similar response for the 1c compound was observed with the 18 nM reference agonist of DAMGO (E_max_% = 35 and pK_b_ 8.05). All the results are shown in Table 4. Together with the tested compounds, the reference compounds were included in the same assay, whose results are consistent with those reported in assay data sheets and in the literature [17,18]. 

**β-Arrestin Recruitment.** Since the recruitment of β-arrestin constitutes a signal transduction pathway independent of the G protein, studies on the effect of the compound on the recruitment of β-arrestin were carried out. The results are presented in Table 5. There were no effects of the new compounds on β-arrestin recruitment in either the agonist or antagonist assays. A summary of data on the recruitment of β-arrestin is presented in the Appendix A. 

## 4. Conclusions

The lack of orthosteric affinity and the inhibition efficacy of DAMGO suggest that 1-(4-methylphenyl)-2,3-dihydro-1*H*-imidazo[1,2-*a*]imidazole-5,6-dione is a negative allosteric modulator of the human MOP (NAM). The biological evaluation reveals **a** different mechanism of activation than that characteristic for the classical opiates; therefore, further investigations are required to determine its accurate effect on the MOP, especially the mechanism of enhancement of the action of typical orthosteric MOP ligands. 

## Data Availability

Not applicable.

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
