# Peer review of "Modulation of the MOP Receptor (μ Opioid Receptor) by Imidazo[1,2-a]imidazole-5,6-Diones: In Search of the Elucidation of the Mechanism of Action"

_molecules, 2022, doi:10.3390/molecules27092930_

Round 1
Reviewer 1 Report
The authors of the article synthesized a series of derivatives of 1-aryl-5,6(1H)dioxo-2,3-dihydroimidazo[1,2-a]imidazole and studied their activity on human mu-opioid receptors (MOP). Indeed, the modus operandi of all opioids utilized in clinical practice nowadays is based, at least partially, on modulation of the MOP receptor, with some types of opioids possessing an ancillary activity at other opioid receptors or receptors different from the opioid family. Employment of opioids is also associated with the heavy side effects and their abusive usage. That is why the development of new OP3 modulators is an important direction in the chemistry of opioid analgesics.
This is the solid work with the scientific content of good quality. However, there are several important aspects which should be modified:
- Description of the current state-of-the-art in the introduction is unsatisfactory. There exist a plethora of new studies and reviews on the MOP modulators in the last 10 years, which were not mentioned.
- The acronyms - DAMGO, MOP, DOP, KOP, CTOP, CTAP, OD, etc. - should be deciphered once when they are used for the first time.
- Scheme 1 should include the nomenclature of the studied complexes: 1a, 1b, 1c, etc.
- If the absorbance spectra at Figure 8 are cut-off at OD higher than 3.5, the explanation should be explicitly included in the caption.
- Number of Figures should be reduced. Less significant Figures should be transferred to SI.
- Conclusion should be expanded.
- English grammar should be improved. Punctuation should be improved, a comma should not separate a subject and a verb. There exist no noun "derives" (in introduction) or the verb "determinate" (in conclusion).
In my opinion, after these corrections are implemented, the manuscript may be published in Molecules.
Author Response
Answer to the first Referee comments:
We thank referee for diligent work. We agree with all comments and change the text accordingly to recommendations.
Introduction.
Additional references especially presenting new discoveries in the field of MOP ligands were added (Ref. 8-12).
Presentation of the results.
In chemical part scheme and general procedure was left in the main text. Physicochemical and spectral data were moved to the Supplementary Materials. In pharmacological part only tabularized results were left in the main text. Other results and figures were moved to the Supplementary Materials.
Acronyms.
Part of acronyms has been explained at the beginning of the manuscript and others once they appear in the text for the first time.
Scheme 1.
The codes for all derivatives were added to the Scheme 1.
Figure 8.
Cutting spectrum at 3.5 OD value was due to sharpness of the signals. Presentation of the full spectrum would make presented differences invisible. Such explanation was added to the caption of the figures. Those figures were moved to the Supplementary Materials.
Conclusion.
It was expanded by adding information concerning activity of all compounds in the biological tests.
Grammar.
Text was carefully checked by native speaker and all mistakes in grammar and wording were fixed.

Reviewer 2 Report
I have the following points to make.
- In Scheme 1. Chemical structure should b numbered (1,2,3, and 4)
- NMR coupling constants (J values) are missing. For example, splitting patterns are mentioned as dd, but no coupling constant values are provided.
- Be consistent with compound names. 1a, 1b… or 1A, 1B
- Why figures and tables are not referenced in the text.
- Figure caption should be more informative.
- Results and material and methods are not segregated properly.
- Abstract should be single paragraph.
Author Response
Answer to the first Referee comments:
We thank referee for diligent work. We agree with all comments and change the text accordingly to recommendations.
Introduction.
Additional references, especially presenting new discoveries in the field of MOP ligands were added (Ref. 8-12).
Presentation of the results.
In chemical part scheme and general procedure was left in the main text. Physicochemical and spectral data were moved to the Supplementary Materials. In pharmacological part only tabularized results were left in main text. Other results and figures were moved to the Supplementary Materials.
Scheme 1.
The codes for all derivatives were added to the Scheme 1 and they were unified within the text and Supplementary Materials.
NMR.
Coupling constants for H NMR spectra were added as well as H, C and MS spectra in Supplementary Materials.
Compounds nomenclature.
All compounds in the whole text were described by respective codes.
Figures.
Number of figures in the main text was reduced and other were moved to the Supplementary Material. Captions were extended to be more informative.
Results and Material and Methods.
They were segregated properly. Proper referring to the Supplementary Materials in the text were introduced.
Abstract.
Abstract was formatted to the single paragraph.
Grammar.
Text was carefully checked by native speaker and all mistakes in grammar and wording were fixed.

Reviewer 3 Report
The manuscript entitled “Modulation of the OP3 receptor (MOP) by imidazo[1,2-a]I midazole-5,6-diones. In search for solving the mechanism of action” by Dominik Straszak et al. describes the synthesis of a few 1-aryl-5,6(1H)dioxo-2,3-dihydroimidazo[1,2-a]imidazole derivatives their activity on human μ-opioid receptors. Overall, the manuscript can not be published in the current form and requires major corrections. In addition, I have the following comments and suggestions for the authors:
- The manuscript contains orthographic mistakes and should be corrected (marked by the yellow color).
- The general synthetic procedure is not full and requires corrections. According to the literature data, the boiling point of diethyl oxalate is 185 oC. “The course of the reaction was measured by the amount of the collected ethanol” by Dean-Stark apparatus? “The mixture was cooled for the next 24 hours” and obtained solid was filtered off or the solvent was evaporated under vacuum, etc.?
- Please, add coupling constants for the doublet of doublets in 1H NMR spectra.
- 13C NMR spectra are missing for the compounds.
- Please, indicate the purity of synthesized compounds.
- Supplementary materials are missing. High-quality figures of 1H and 13C NMR and MS spectra for all compounds should be added.
- Figures 1-3,6 are hard to read. Please, add a color for each curve.
- The Reference list is not formatted correctly and should be corrected according to the rules of the Journal.

Author Response
Answer to the first Referee comments:
We thank referee for diligent work. We agree with all comments and change the text accordingly to recommendations.
Grammar.
All marked mistakes were corrected. In addition text was read by native speaker and corrected accordingly to recommendations.
General synthetic method.
Description of the synthetic methodology was changed accordingly to the referee suggestions. All physicochemical and spectral properties of compounds obtained were moved to the Supplementary Materials. Purity of compounds obtained was also added.
Spectra.
Coupling constants for H NMR spectra were added. The H and C NMR as well as MS spectra were also added to the Supplementary Materials. Supplementary Materials were prepared and added to the manuscript.
Figures 1-3, 6.
Those Figures were improved to enhance the visual effect. Figures were moved to the Supplementary Materials.
References.
References were formatted accordingly to the Molecules requirements. Five references concerning recent discoveries on MOP ligands were added (Ref. 8-12).
Other.
DMSO-d6 nomenclature for deuterated dimethyl sulfoxide is commonly used by suppliers.
Nomenclature for DMSO was unified in the text and chemical part.
Information concerning purity of all compounds was introduced to the description of the general procedure.

Round 2
Reviewer 2 Report
The authors have addressed the all concerns raised by the reviewers.
Author Response
Thank you very much for reviewing our manuscript and for your comments. Thank you also for accepting changes we applied.
Reviewer 3 Report
The authors improved the quality of the manuscript, and all my previous comments were taken into account. In my view, this manuscript could be accepted in current form.
Author Response

(The authors gave the same response as above.)
